# Breeding Wheat for Powdery Mildew Resistance: Genetic Resources and Methodologies—A Review

Theresa Bapela [1,2,*], Hussein Shimelis [1], Tarekegn Terefe [2], Salim Bourras [3], Javier Sánchez-Martín [4], Dimitar Douchkov [5], Francesca Desiderio [6] and Toi John Tsilo [2]

1 African Centre for Crop Improvement, University of Kwa-Zulu Natal, Private Bag X01, Scottsville, Pietermaritzburg 3209, South Africa; shimelish@ukzn.ac.za
2 Agricultural Research Council-Small Grain Institute, Bethlehem 9700, South Africa; terefet@arc.agric.za (T.T.); tsilot@arc.agric.za (T.J.T.)
3 Department of Forest Mycology and Plant Pathology, Swedish University of Agricultural Sciences, Almas Alle 5, 75007 Uppsala, Sweden; salim.bourras@slu.se
4 Department of Microbiology and Genetics, Institute for Agrobiotechnology Research (CIALE), University of Salamanca, 37008 Salamanca, Spain; j.sanchezmartin@usal.es
5 Research Group Biotrophy and Immunity, Leibniz Institute of Plant Genetics and Crop Plant Research (IPK), 06466 Gatersleben, Germany; douchkov@ipk-gatersleben.de
6 Council for Agricultural Research and Economics-Research Centre for Genomics and Bioinformatics, Via S. Protaso 302, 29017 Fiorenzuola d'Arda, Italy; francesca.desiderio@crea.gov.it
* Correspondence: ttbapela@gmail.com

**Abstract:** Powdery mildew (PM) of wheat caused by *Blumeria graminis* f. sp. *tritici* is among the most important wheat diseases, causing significant yield and quality losses in many countries worldwide. Considerable progress has been made in resistance breeding to mitigate powdery mildew. Genetic host resistance employs either race-specific (qualitative) resistance, race-non-specific (quantitative), or a combination of both. Over recent decades, efforts to identify host resistance traits to powdery mildew have led to the discovery of over 240 genes and quantitative trait loci (QTLs) across all 21 wheat chromosomes. Sources of PM resistance in wheat include landraces, synthetic, cultivated, and wild species. The resistance identified in various genetic resources is transferred to the elite genetic background of a well-adapted cultivar with minimum linkage drag using advanced breeding and selection approaches. In this effort, wheat landraces have emerged as an important source of allelic and genetic diversity, which is highly valuable for developing new PM-resistant cultivars. However, most landraces have not been characterized for PM resistance, limiting their use in breeding programs. PM resistance is a polygenic trait; therefore, the degree of such resistance is mostly influenced by environmental conditions. Another challenge in breeding for PM resistance has been the lack of consistent disease pressure in multi-environment trials, which compromises phenotypic selection efficiency. It is therefore imperative to complement conventional breeding technologies with molecular breeding to improve selection efficiency. High-throughput genotyping techniques, based on chip array or sequencing, have increased the capacity to identify the genetic basis of PM resistance. However, developing PM-resistant cultivars is still challenging, and there is a need to harness the potential of new approaches to accelerate breeding progress. The main objective of this review is to describe the status of breeding for powdery mildew resistance, as well as the latest discoveries that offer novel ways to achieve durable PM resistance. Major topics discussed in the review include the genetic basis of PM resistance in wheat, available genetic resources for race-specific and adult-plant resistance to PM, important gene banks, and conventional and complimentary molecular breeding approaches, with an emphasis on marker-assisted selection (MAS).

**Keywords:** adult-plant resistance; *Blumeria graminis*; marker-assisted selection; race-specific resistance; *Triticum aestivum* L.

## 1. Introduction

Wheat (*Triticum aestivum* L.) is an important commodity crop that provides food to about 30% of the world's population and accounts for over 20% of human-consumed calories [1]. Over the last decade, global wheat production has shown an increasing trend except for a slight decrease during the 2018/2019 growing season [2]. The recent Ukraine/Russia crisis has significantly highlighted the dependence of most African countries on external resources including fossil fuels and grain wheat originating from these two countries. As a result, there are now challenges to acquiring wheat, fungicide, and fertilizer from external markets. This has further affected many other components of the food supply chain [3]. Furthermore, the combined interplay of these factors has measurable negative impacts on food security.

It is worth noting that the global human population is expected to increase to 9 billion by 2050 [4] increasing the global demand for food. Current wheat yield gains are estimated at around 0.5 to 1% per annum, below the 2.4% required to meet the global demand for this commodity [5,6]. Consequently, wheat production should increase by up to 70% to meet the projected global demand for wheat products by 2050 [7,8]. The average yield of wheat has been stagnant by up to 40% in recent years, which shows that the current output and productivity rate are not sufficient to ensure future food security. The shortage of arable land, the tension on water resources, and climate change limit the potential to expand production areas to increase output. Furthermore, the low productivity of wheat is also attributed to several biotic and abiotic factors that reduce its yield potential [9,10]. Therefore, new-generation wheat cultivars need to be developed with enhanced tolerance/resistance to a plethora of stresses, e.g., resistance to diseases, pests, soil alkalinity and salinity, and nitrogen use efficiency to enhance yield potential.

Diseases such as powdery mildew (PM), caused by the fungal pathogen *Blumeria graminis*, are widespread globally and have contributed to significant yield losses [11]. The genus *Blumeria* is monophyletic, i.e., it includes only one species "*Blumeria graminis*", which is subdivided into eight *forma speciales* that infect grasses and cereal crops including wheat, barley, oats, and rye [12]. Wheat (*sensu lato*) can also be infected by *B.g. dicocci* (tetraploid durum wheat) as well as *B.g. triticale*, which is a hybrid between wheat and rye mildew with an expanded host range that can infect triticale and wheat [13]. The effect is major because breeding for PM resistance in wheat makes no distinction between these *formae speciales* and their prevalence on different cultivars and in different regions is largely unknown. Thus, developing powdery mildew-resistant cultivars based on a better understanding of mildew populations and the interplay between adapted and non-adapted *forma speciales* should lead to improved strategies in identifying new and novel genetic sources of resistance against PM.

The lack of progress in developing and deploying resistant cultivars can be attributed to several factors, including the difficulties encountered in screening (i.e., PM screening is more complex than expected), the poor understanding of the genetic basis of disease resistance, and the polygenic nature of the resistance that is highly influenced by environmental conditions [14]. Identifying genetic variation PM resistance is an important preliminary step to developing resistant cultivars. Selection for resistance must first be carried out on a large panel of germplasm in different sites. The expression of PM resistance is highly variable across sites and seasons, which makes it difficult to ensure consistent and discriminatory disease pressure which could confound the selection and identification of highly resistant genotypes [15,16]. Resistance phenotype is due to resistance genes that are inherited from one generation to another (parent-to-offspring relationship). In countries where wheat PM epidemics have been recently reported, virulence frequencies of the races/isolates to the newly reported resistance genes are generally lower. For example, in South Africa (SA), the identified PM isolates were mostly avirulent to the newly reported *Pm* genes (*Pm25-Pm53*) except *Pm35* [17]. In general, there are no/few genes that confer resistance to all pathogen races. For this reason, tentative and short-lived genes for powdery mildew resistance have been identified but their use in developing resistant cultivars has been minimal due to a

lack of durability [18,19]. However, the pyramiding of multiple sources of resistance genes has been suggested as the most effective strategy to increase the durability of resistance against most fungal diseases in wheat, a strategy highly impossible to achieve, through conventional breeding methods.

Complementing conventional breeding with molecular techniques has the potential to increase selection efficiency for PM resistance and yield-related traits. This is because molecular markers are not influenced by environmental variability and could increase understanding of the genetic basis of PM resistance. Over the last few decades, advances in genomic-assisted breeding and the application of Next-Generation Sequencing (NGS)-based genotyping technologies have contributed mainly to accelerating the identification and introduction of PM resistance traits into commercial cultivars. More than 240 PM resistance genes/loci have been reported on all 21 wheat chromosomes, with more than 60 genes/alleles identified and located on 18 chromosomes [20–22]. Of these, the "A" subgenome (1A, 2A, and 7A) and B subgenome (2B, 5B, and 6B) have been shown to encode several major PM resistance genes [21,23]. Furthermore, 19 PM resistance genes/alleles were cloned, e.g., *Pm1a*, *Pm2*, *Pm3* (*Pm3a* to *Pm3g*, *Pm3k*, and *Pm3r*), *Pm4*, *Pm8*, *Pm17*, *Pm21*, *Pm24*, *Pm38/Yr18/Lr34/Sr57*, *Pm46/Yr46/Lr67/Sr55*, *Pm60* and WTK4 [16,24–33]. Of the 19 clones, only *Pm3k* was isolated from tetraploid wheat [34]. More than half of these genes were introgressed from wheat progenitors and related species. Despite this, they have not been widely commercially deployed due to their suppressed resistance levels and association with negative linkage drag [35,36]. For example, rye translocations have been associated with bad dough traits [37,38]. Some of these race-specific resistance genes exhibited a "boom-and-bust" cycle due to the emergence of new virulent races [15,18], thus exerting strong selection pressure. Therefore, identifying PM resistance genes in common wheat, including landraces, would be more beneficial for developing cultivars with good agronomic performance and minimum linkage to deleterious traits [19,39]. Therefore, the objectives of this review are to outline the status of breeding for powdery mildew resistance, present the pathogenesis and distribution of PM isolates, important gene banks and databases, available genetic resources, as well as complimentary breeding approaches in developing powdery mildew-resistant cultivars and provide an outlook on the way forward.

## 2. Constraints to Wheat Production

Wheat production in Africa is insufficient to sustain the growing population thus increasing most countries' dependency on imports for inputs such as fertilizers, fungicides, pesticides, herbicides, and oil, and the recent crisis between Ukraine/Russia has significantly restricted access and movement to these resources. On the other hand, wheat is no longer a profitable crop in most production regions, thus farmers are transiting to more profitable crops such as maize and soybean. In addition to reliance on inputs, wheat production, and productivity are constrained by additional factors such as insufficient arable land, low-yielding cultivars, soil infertility, drought, diseases, and pests which collectively reduce the on-farm yield [10,40–47]. Of these, diseases are the most prominent constraints impacting wheat yield. Around 200 diseases and pests have been reported on wheat, of which 50 are economically devastating to farmers' crops [48]. In 2019, global yield losses from diseases and pests in wheat were estimated at 22% [47]. Of these, PM has been the most prevalent and destructive disease threatening small grain production such as wheat and barley, and to a smaller extent in oats and rye [49–52].

Wheat powdery mildew has shown significant global incidents over the last four decades [53]. The disease is ranked sixth out of the 10 most important fungal diseases of wheat [54] and the 8th highest yield loss contributor of wheat globally [55]. Dense cultivation associated with the use of semi-dwarf cultivars and high levels of nitrogen application favors disease development and severity [56]. Temperatures below 25 °C and relative humidity levels of ≥50% are optimum conditions for the development and spread of the pathogen. Important characteristics of powdery mildew that enable rapid dissemination

and adaptation include a short life cycle, airborne spores that can easily travel over long distances, and the rapid evolution of new virulent races. In alignment with the above-mentioned characteristics and climatic conditions, the cooler to humid regions in Asia, Africa, Europe, and the United States favor the development of the pathogen [51,52,57,58]. In response to mitigating the disease, breeding for race-specific, quantitative, and adult-plant resistance as well as fungicide application and the key control strategies for PM in cereals including wheat. Thus, while using resistant cultivars is considered one of the most effective and environmentally friendly strategies to mitigate PM and reduce the application of fungicides, the rapid evolution of new virulent races can severely reduce the durability of genetic resistance in the field. For example, the ineffectiveness of resistance genes *Pm3a*, *Pm4a*, and *Pm17* has been reported in the USA "Mid-Atlantic States" [15], *Pm8* in China [59] and *Pm1*, *Pm3*, *Pm4*, and *Pm5*, *Pm7*, *Pm24* and *Pm28* in Australia [18]. From the 346 *B.gt* isolates derived from six countries, all the countries had the highest virulence frequencies for genes *Pm6*, *Pm8*, and *Pm17* with additional different genes for each country. This includes *Pm1a*, *Pm35* and *MIUM15* to Egyptian isolates; *Pm1a*, *Pm4a* and *Pm4b* and *MIUM15* to Turkish isolates; *Pm2*, *Pm4a*, *Pm4b*, *Pm25* and *Pm35* to Russian/Kazakhstan isolates; *Pm2*, *Pm3a*, *Pm3b*, *Pm4a*, *Pm25*, *Pm34*, *Pm35* and *NCA6* to Unites states isolates; *Pm1a*, *Pm2*, *Pm3a*, *Pm3b*, *Pm4a*, *Pm4b*, *Pm34* and *NCA6* to Brazil isolates; *Pm1a*, *Pm3b*, *Pm35* and *MIUM15* to Australian isolates and lastly, *Pm3b*, *Pm34* and *Pm35* to South African PM isolates [17]. Therefore, the identification or development of new resistance sources to new pathogen races is needed to achieve an ongoing effort to control the pathogen. However, to achieve a more effective and durable resistance, a gene pyramiding strategy with different *Pm* genes could serve as a sustainable way of exploiting resistance genes from elite/novel genetic resources [60–63]. Furthermore, the replacement of ineffective cultivars and diversification of resistance sources carrying several resistance genes is paramount to maintaining a healthy crop [64].

## 3. Pathogenesis, Distribution, and Economic Importance

### 3.1. Pathogenesis

### 3.1.1. Life Cycle and Epidemiology

Powdery mildew can reproduce both asexually and sexually, which leads to the production of asexual conidial and sexual ascospores, respectively [65]. The most important infections are initiated by the release of sexual ascospores from the fruiting bodies, called chasmothecia, infecting crops grown in autumn and spring. Sexual ascospores usually appear in asci within 3 to 5 days of moisture contact/exposure. Secondary infections involve the emergence of germ tubes that elongate and differentiate into a structure, called an appressorium. After 6 days, hyphae differentiate to form a conidial structure called conidiophores, which matures between 8 and 10 days [50]. Typical mildew colonies start as small whitish round spots which can be surrounded by chlorosis and later become tan or brown. As the lesion ages, mycelium becomes dense and turns grey on leaves and heads.

Powdery mildew thrives well under high relative humidity (50–100%) and low temperatures ranging from 15 to 25 °C, as temperatures of more than 25 °C delay the development of the disease [66,67]. The PM outbreaks during the growth season occur during conditions of alternating winter, spring, and summer with some wind to ensure effective dissemination of the conidia. Wheat PMs are host specific and can only grow on one host species with the only exception of *B.g. triticale* [13]. The fungus survives on wheat hosts mainly as dormant mycelium or conservation structures (chasmothecia). The primary infections involve chasmothecia (135 to 224 μm in diameter) produced during the late spring or summer in the mycelium, which are resistant to extreme weather conditions and to moisture loss, thus serving as an important survival mechanism and source of inoculum for the next season [68]. Rising temperatures (3–31 °C) with an optimum of 15 °C and ~100% relative humidity) in the spring enable dormant mycelium to start growing and rapidly producing conidia. The overwintering of chasmothecia and over-summering of mycelium and conidia allows the pathogen to survive adverse periods [50]. The disease may have a

devastating impact on grain yield and quality [69,70], while severe infections may result in leaf death [52].

### 3.1.2. Damages

Contrary to nectotrophs that kill the host cells, PM is an obligate biotroph, highly dependent upon a live host plant to complete its life cycle and cause damage [54]. Favorable conditions enable the disease to cover the upper leaf surface, thus withering and weakening the host. PM symptoms are commonly found on the lower oldest leaves and progress with the plant growth damaging the upper leaves, heads, and awns of susceptible cultivars [71]. Shady low areas that trap damp air, and places with high plant density and poor air circulation favor the development of this disease.

The PM infections occurring during the seedling stage hinder the growth and development of wheat where plants may die due to severe infections. Furthermore, infections occurring at the tillering stage could inhibit the development of wheat roots and reduce the formation of tillers. Moreover, infections during the heading and flowering stages can decrease the number of grains per ear, grain filling, and weight [72–75]. Overall, PM epidemics may result in reduced grain yield and yield-related traits (number of tillers, kernel number per head, kernel weight, grain numbers, grain filling), and losses in grain quality thus affecting end-use quality parameters (such as wheat processing, milling, baking quality) [69,71,72,76–80]. The conversion of sugar to starch in the wheat kernels was suppressed by PM at the early infection stage while at the later infection stage, there was an adequate substrate for starch synthesis in susceptible cultivars [81].

### 3.1.3. Population Genetics

Knowledge of the population genetics of plant pathogens is essential for a full understanding of the disease ecology, epidemiology, and evolutionary and genetic trajectory to effectively deploy host-plant resistance and agrochemicals, and ultimately control the plant pathogen [82]. Population genetics involves the genetic and evolutionary processes such as mutation, genetic drift, migration/gene flow, natural selection, recombination, and mating systems that collectively cause the genetic change or the evolution in populations under the influence of hosts, pathogens, environment, agricultural practices, and human activities [83,84]. These evolutionary forces determine the pathogen's adaptability to inconsistent environmental conditions such as fungicide resistance or overcoming a resistance gene in the plant host thus causing considerable farm-level losses. Mutation is the primary source of pathogen genetic variation and adaptation to new environments. A high mutation rate enables the pathogen population to adapt rapidly to new resistance genes or fungicide application [84]. However, it becomes short-lived once the adaptation has been successfully achieved at such rates. Genetic drift involves inadvertent random events influencing allele frequencies of the pathogen population [84]. For example, the wheat growing season is short, meaning a deprivation of food sources for the pathogen when the season ends, resulting in genetic drift. Migration/gene flow entails the exchange of genetic information of genotypes from one location to another, introducing novel alleles/gene combinations from bordering pathogen populations thus increasing the genetic variation. Natural selection is the prominent source of genetic variation and the evolutionary trajectory of pathogens. The phenomenon is intensified by modern production systems that routinely practice monoculture. Directional selection for a trait of interest is behind the loss of effective resistance genes/alleles in most cultivars as well as fungicide resistance [82]. Recombination and mating systems involve the independent assortment of DNA sequences between the same or different genomes, either through sexual recombination or hybridization/horizontal gene transfer i.e., heterothallic or homothallic [85].

### 3.2. Geographic Distribution and Economic Importance

Directional selection increases the frequency of the virulent pathotypes then spread to bordering areas or countries through natural or human selection i.e., mediated gene

flow [81]. Wheat powdery mildew is widely distributed in regions of temperate, cool to humid climatic conditions such as Asia, Europe, Africa, the United States of America, and Oceania [51,53,86–89]. In recent decades, the pathogen populations have spread intensely to warmer and drier areas/regions as a result of modern production systems i.e., dense cultivation, over-irrigation, and high levels of nitrogen fertilization [56,89]. According to Morgounov et al. [52], PM disease outbreaks have been reported in 51 countries during 1969–2010 amounting to 1047 reports ranging from 31 to 83% with a global dominance of 54%. In Africa, the disease has been reported in 17 countries including the Western Cape of South Africa and bordering countries [17,90]. Estimating the economic threshold from the wheat PM epidemics can be challenging especially since disease development is dependent upon yearly climatic conditions (season vs. temperature and rainfall). The cultivar, location, and land area planted also determine the incidence and severity of PM infections [67,69,91,92]. In wheat, the disease greatly impacts grain yield by reducing the number of heads, kernel size, and weight; the number of productive heads and tillers [69,70,92,93]. Yield losses from PM have reached up to 23% in Egypt, 35% in Russia, 40% in China, 50% in Denmark, and 62% in Brazil [11,51,93]. The highest yield losses have been reported in Central and Eastern Europe (72%) and Western and Southern Europe (93%) [53]. According to Tang et al. [58], the percentage of affected wheat-producing regions in China has increased by 8.5% per decade from 1970 to 2012. Approximately, 8 million hectares were infected with PM over the last decade in China [42]. These incidents indicate that PM is re-emerging as a global food security threat. Therefore, global wheat-breeding programs should prioritize preventative, effective, and environmentally friendly methods to control the disease. Understanding factors influencing PM resistance breeding including the etiology, pathogen and its virulence mechanisms/spectrum, the host and its resistance mechanisms as well as the environmental factors favoring pathogenesis is essential for effective control of PM.

## 4. Current Control Strategies

The occurrence, development, and severity of diseases are often determined by the presence of a susceptible host, a virulent pathogen, and a conducive environment. This is referred to as a disease triangle. Therefore, the need for intervention strategies to mitigate the disease is paramount. Several control strategies for powdery mildew are available including cultural, biological, chemical, and genetic resistance [10,90]. However, due to limited studies on cultural and biological control, chemical control and host-plant resistance are widely used against PM and other foliar diseases globally and in Africa including SA [56].

### 4.1. Monitoring: Remote Sensing Technologies

Powdery mildew negatively impacts wheat growth, development, production, and productivity. Thus, correct timing and monitoring of the disease is paramount for preventing considerable farm-level yield losses. Conventionally, PM is recognized by white, fluffy colonies on the wheat leaf surface [90]. The advent of remote sensing technologies has allowed researchers to routinely monitor crop stands and detect an array of diseases in crops, on a large germplasm collection, within a short space of time, consequently complementing conventional methods [94]. This includes spectral sensors, hyperspectral imaging, chlorophyll fluorescence, and thermography [95,96]. For example, Figure 1A,B depicts PM colonies and feeding structures on a wheat host revealed by a microscope. On the other hand, high-throughput phenotyping techniques such as machine-learning (ML)-aided phenotyping, Macrobot 2.0 for multimodal imaging, Zeiss AxioScan.Z1 high-performance microscopy slide scanner and Convolutional Neural Network (CNN)-aided analysis) have allowed digital measurement of the disease, increasing the accuracy of quantifying the leaf area affected by the disease (Figure 1C–H). These technologies are timely, fast, and non-destructive for precise early detection and pathogenesis monitoring of PM and estimating grain yield [97–99]. Furthermore, the biochemical and physiological

status of the plant is easily and effectively determined through the combination of images and spectrums [94,99]. This includes pigments such as chlorophyll necessary for photosynthesis, carotenoids for plant survival through photosynthetic and nutritional functions, and anthocyanin for plant physiology [100]. Different plant–pathogen interactions influence the spectral signature (spectral reflectance pattern). Changes in the spectral pattern and intensity are used to derive the histological and physiological/biological status of the plant–pathogenesis–environment interaction [96]. According to Feng et al. [101], different host species and pathogens show variability in spectral traits thus producing varying waveband reflectivity in response to the disease. For example, Figure 1.

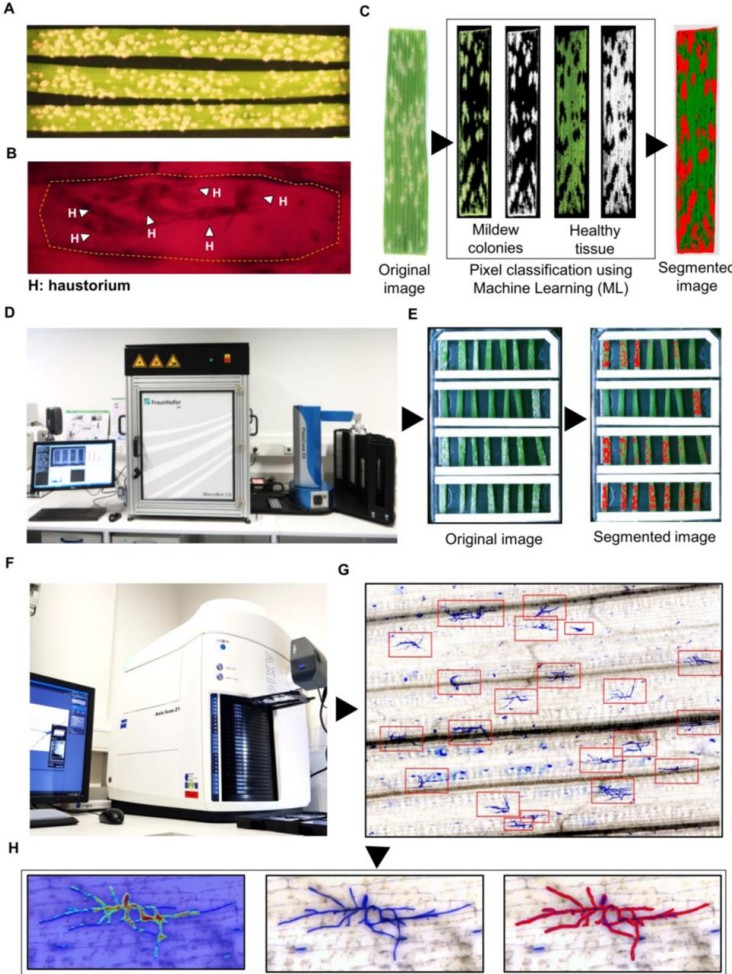

**Figure 1.** Next-generation machine-learning (ML)-and artificial-intelligence (AI)-aided phenomics for precision breeding of powdery mildew resistance in wheat. (Photos A to C supplied by Salim Bourras and D to H by Dimitar Douchkov). (**A**) Macroscopic "powdery mildew" colonies from the genome reference isolate *Bgt* _96224 growing on the susceptible hexaploid wheat cv Chinese Spring. (**B**) Multiple mildew intracellular haustorial feeding structures colonizing a wheat epidermal revealed by light microscopy. (**C**) Machine-learning (ML)-aided phenotyping of leaf coverage by wheat powdery mildew colonies using a pixel classification approach. (**D**) Macrobot 2.0 fully automated high-throughput multimodal image acquisition system at IPK (Germany). (**E**) An example of ML-aided feature extraction based on pictures taken with Macrobot 2.0. (**F**) Zeiss AxioScan.Z1 high-performance microscopy slide scanner allowing fully automated microphenomic acquisition at IPK (Germany). (**G**) Convolution-al Neural Network (CNN) aided analysis of powdery mildew micro-colonies at 48-h after infection with (**H**) visualization of calculated probability for the presence of fungal structures and marked micro colony area. Original pictures in panels (**A**–**C**) are courtesy of co-author Salim Bourras. Original pictures in (**D**–**H**) are courtesy of co-author Dimitar Douchkov.

*4.2. Intervention Strategies*

4.2.1. Integrated Management Strategies

Ensuring excessive planting of resistant wheat varieties over susceptible ones can slow the pathogen rate and disease progression while minimizing the reliance on fungicides in mitigating the disease. Late planting can delay plant growth and ultimately the conditions that favor (warm and damp periods) the development of the pathogen. Though this might be effective in reducing powdery mildew rates, it costs profit to farmers and growers since the planting of late-growing cultivars is sometimes associated with reduced yield potential. Excessive nitrogen fertilization favors disease development and thus should be kept optimum. Practicing crop rotation can also help reduce the inoculum levels from season to season [56,102].

4.2.2. Chemical Control

To control the disease, the application of foliar fungicides is significantly recommended; however, depending on locality, timing, disease pressure, and host resistance level, the yield responses can vary [91,103]. For example, a trend in fungicide application to control foliar diseases has revealed a fluctuating but increasing pattern between 1995 and 2010 in Ethiopia [10]. Furthermore, nearly 500 tons of contrasting fungicides (active ingredients) were used during this period [10]. To date, only a few specific fungicides are available for mitigating PM. Some of the effective chemical fungicides used for seed treatment and foliar application include carbendazim, demethylation inhibitors, carboxin, quinone outside inhibitors, methyl benzimidazole carbamates, thiram and metalaxyl [10,47]. However, the use of fungicides is not environmentally and economically friendly as it poses a threat to human and animal health and increases production costs. For example, the current EU initiative concerning the prohibition of using chemical pesticides [104], forces researchers and breeders to develop and routinely practice alternative ways of mitigating crop diseases. Most farmers worldwide are reluctant to step outside their comfort zone and therefore still plant wheat varieties introduced since the green revolution either due to poor seed distribution/circulation and poor farmer adoption or uptake of newly released varieties as well as a preference for traditional varieties over modern poorly adapted varieties. Their variety choice relies on traditional knowledge and past farming experiences hence knowledgeable about traits adapted to diseases and pests among other traits. Therefore, the application of agrochemicals on genetically diverse pathogen populations, which occur on cultivars grown repeatedly, can render them ineffective. In addition to the lack of replacement of old seed varieties, ineffective/repeated use of the same fungicides and reliance on only a few active substances leads to the development of fungicide resistance in pathogen populations [10,105]. Farmers, researchers, and breeders can devise strategies such as correct timing and accurate application of fungicides at the target plant growth stage to effectively reduce the incidence and severity of powdery mildew and other foliar diseases in wheat and other crops [56,106]. For example, the application of a single fungicide resulted in an average of 8% grain-yield response from six trials in Australia. However, multiple applications of the fungicides at the correct growth stage of the plant doubled the grain-yield response by up to 20% [107]. For example, in Sidney, the application of foliar fungicide at the flag leaf stage in the spring season significantly reduced disease severity by up to 84%, increasing grain protein content, grain volume weight, seed weight, leaf area, grain yield, and leaf greenness, resulting in economic returns of up to USD 204 ha$^{-1}$ [106]. Rotation between fungicides from different chemical groups can also limit the development of fungicide-resistant strains [56].

4.2.3. Host-Plant Resistance

The use of cultivar mixtures and resistance gene pyramiding are two well-documented approaches to genetically control wheat powdery mildew [56,61,62]. In particular, gene pyramiding remains the most feasible, environmentally and economically friendly means of controlling *B.gt* and ensuring durable resistance [63,108–110]. This is important, especially

as the use of a single gene is ephemeral due to evolving PM virulent races. For example, only a limited number of PM resistance genes are still effective including *Pm2, Pm3a, Pm3e, Pm4a, Pm13* and *Pm27* to Western Australia isolates [18], *Pm3* alleles (*Pm3a, Pm3b, Pm3c, Pm3d,* and *Pm3f*) [19], *Pm6* [111] and *Pm2, Pm4b* and *Pm8* [112] and *Pm1a* in SA [17]. In a recent study, 45% of the 15,944 bread wheat genotypes screened were resistant to PM isolate in India [113]. In the West Siberian region, adult-plant resistance (APR) was observed in only 6% (six out of 97) of varieties screened with mildew isolates from the region thus representing a small portion of effective PM resistance genes [114]. Furthermore, only 5% (59 of 1297) of landraces exhibited resistance to PM in the US [19].

## 5. Host-Plant Resistance: Progress and Achievements

### 5.1. Resistance Types for Powdery Mildew

Plants use diverse mechanisms against pathogens e.g., race-specific, non-race-specific, qualitative, and quantitative resistance and the genetic status of both (the host plant and the pathogen race) determines the consequence of this host-pathogen interaction. Table 1 presents reported race-specific and race-non-specific genes for wheat powdery mildew. Over recent decades, most research studies have focused on major *Pm* resistance genes presumed to be race-specific or qualitative. For example, the *Pm3* resistance gene (and its alleles) is widely explored since it is simply inherited, transient, and easy to manipulate and express complete resistance which is often associated with the hypersensitive response which is effective against a few pathogen races and can be easily defeated by new virulent pathogen races [24,62,115]. Until recently, adult-plant resistance has been the focal point of most studies, as it is associated with non-race-specific resistance as well as durable resistance which involves the interplay of multiple genes that delay and reduce the infection, growth, and reproduction of the fungus at the adult-plant stage [116]. Nevertheless, more than 240 PM resistance genes were identified on all 21 wheat chromosomes even though they were not evenly distributed on each chromosome.

**Table 1.** Genes associated with powdery mildew race-specific and race-non-specific resistance, germplasm source, and their references.

| Reported Genes | Germplasm Source | References |
|---|---|---|
| Race-specific resistance | | |
| *Pm2* | *A. squarrosa* | [117] |
| *Pm3a-pm3j* | *T. aestivum* L. | [24] |
| *Pm4* | *T.aestivum* L. | [31] |
| *Pm4b, 4c* | *T. aestivum* L. (RE714) | [118] |
| *Pm5* | *T aestivum* L. | [119] |
| *Pm5a* | *T. aestivum* L. | [119] |
| *Pm5b* | *T. aestuvum* L. | [120] |
| *Pm5c* | *T. sphaerococcum* | [120] |
| *Pm5d* | *T. aestivum* L. | [120] |
| *Pm5e* | *T. aestivum* | [121] |
| *Pm8* | *Secale cereale* | [117] |
| *Pm9* | *T. aestivum* L. | [122] |
| *Pm10* | *T. aestivum* L. | [123] |
| *Pm11* | *T. aestivum* L. | [123] |
| *Pm13* | *Aegilops longissima* | [124] |
| *Pm14* | *T. aestivum* L. | [123] |
| *Pm15* | *T. aestivum* L. | [125] |
| *Pm16* | *T. aestivum* L. | [126] |
| *Pm17* | *Secale cereale* | [117] |
| *Pm18* | *T. aestivum* L. | [123] |
| *Pm19* | *A. squarrosa* | [117] |
| *Pm20* | *Secale cereale* | [35] |
| *Pm21* | *Haynaldia villosa* | [127,128] |
| *Pm22* | *T. aestivum* L. | [129,130] |

**Table 1.** *Cont.*

| Reported Genes | Germplasm Source | References |
|---|---|---|
| **Race-specific resistance** | | |
| *Pm23/Pm4c* | *T. aestivum* L. | [131] |
| *Pm24/24b* | *T. aestivum* L. | [132,133] |
| *Pm25* | *T. monococcum* | [134] |
| *Pm26* | *T. turgidum* | [135] |
| *Pm27* | *T. timopheevii* | [136] |
| *Pm28* | *T. aestivum* L. | [137] |
| *Pm29* | *T. aestivum* L. | [138] |
| *Pm30* | *T. turgidum* | [139] |
| *Pm31* | *T. turgidum* | [140] |
| *Pm32* | *Ae. spelltoides* | [141] |
| *Pm33* | *T. turgidum* | [142] |
| *Pm34* | *Ae. tauschii* | [143] |
| *Pm35* | *Ae. tauschii* | [144] |
| *Pm36* | *T. turgidum* | [145] |
| *Pm37* | *T. timopheevii* | [146] |
| *Pm40* | *Elytrigia intermedium* | [147] |
| *Pm41* | *Triticum turgidum* | [148] |
| *Pm42* | *T. turgidum* | [149] |
| *Pm43* | *Thinopyrum intermedium* | [150] |
| *Pm45* | *T. aestivum* L. | [151] |
| *Pm47* | *T. aestivum* L. | [152] |
| *Pm48* | *Ae. tauschii* | [153] |
| *Pm51* | *Thinopyrum ponticum* | [154] |
| *Pm52* | *T. aestivum* L. | [155] |
| *Pm54* | *T. aestivum* L. | [156] |
| *Pm57* | *Ae. searsii* | [157] |
| *Pm59* | *T. aestivum* L. | [158] |
| *Pm60* | *T. urartu* | [159] |
| *Pm61* | *T. aestivum* L. | [160] |
| *Pm63* | *T. aestivum* L. | [161] |
| *Pm65* | *T. aestivum* L. | [162] |
| *Pm66* | *Ae. longissima* | [163] |
| *Pm68* | *T. turgidum* | [164] |
| *Pm69* | *T. turgidum* | [165] |
| *PmCH1357* | *T. aestivum* L | [166] |
| *PmCG15-009* | *T. aestivum* L. | [167] |
| *MG5323* | *T. turgidum* | [135] |
| *MlHLT* | *T. aestivum* L. | [168] |
| *PmG3M* | *T. turgidum* | [169] |
| *MlXBD* | *T. aestivum* L. | [170] |
| *pmHYM* | *T. aestivum* L. | [171] |
| *MIRE* | *T. aestivum* L. | [118] |
| *pmDGM* | *T. aestivum* L. | [172] |
| *pmQ* | *T. aestivum* L. | [173] |
| *PmZ155* | *T. aestivum* L. | [174] |
| *MlLX99* | *T. aestivum* L. | [175] |
| **Race-non-specific** | | |
| *Pm6* | *T. aestivum* L. | [111] |
| *Pm7* | *Secale cereale* | [176] |
| *Pm12* | *Ae. speltoides* | [177,178] |
| *Pm38* | *T.aestivum* L. | [179,180] |
| *Pm39* | *T aestivum* L. | [181,182] |
| *Pm46* | *T.aestivum* L. | [183] |
| *Pm53* | *Ae. speltoides* | [184] |
| *Pm55* | *Dasypyrum villosum* | [185] |
| *Pm56* | *Secale cereale* | [186] |
| *Pm58* | *Ae. tauschii* | [187] |
| *Pm62* | *Dasypyrum villosum* | [188] |
| *Pm64* | *T. turgidum* | [189] |
| *Pm67* | *Dasypyrum villosum* | [190] |
| *pmX* | *T. aestivum* L. | [191] |
| *PmWFJ* | *T. aestivum* L. | [192] |

### 5.2. Pleiotropic APR Genes for Powdery Mildew and Other Wheat Diseases

Varieties with high levels of resistance can be developed by combining or pyramiding multiple race non-specific resistance loci conferring resistance to multiple pathogen pathotypes. This is known as pleiotropic resistance and example of those genes are *Pm38*, *Pm39* and *Pm46* [179,181,193]. This is because race non-specific resistance is commonly associated with lower selection pressure on pathogen populations, a broader spectrum of action, which makes it more durable. Wheat cv Thatcher has been an important donor for APR genes (*Lr34* and *Yr18* for leaf rust and stripe rust, respectively) has been used as a donor parent for many lines including RL6058, RL6077 and 90RN2491 [179,194–198]. The genes on chr 7D, derived from Thatcher, reside in the same region where *Sr57* and *Pm38* [179] are mapped. Furthermore, this region has also been reported to be pleiotropic or linked to wheat spot blotch gene *Sb1* and leaf tip necrosis gene (*Ltn1*) on chromosome 7DS [199]. Therefore, the order of pleiotropism *Lr34/Yr18/Sr57/Pm38/Ltn1*.

The second pleiotropic APR gene *Lr46/Yr29* was found in the International Maize and Wheat Improvement Center (CIMMYT) wheat line Pavon 76 [200] located on chromosome 1BL [201] and has been a major APR source to leaf and stripe rust for nearly half a century. Cultivar Saar has also been a major source for gene *Pm39* for PM resistance for with QTLs detected on chromosome 1BL showing pleitropism to *Lr46/Yr29/Pm39* [180,181]. LTN was also reported pleiotropic or closely linked to the *Lr46/Yr29* locus suggesting an *Ltn* gene [202].

The third pleiotropic APR gene is *Lr67/Yr46/Sr55/Pm46* located on chromosome 4DL which also confers resistance to multiple fungal diseases of wheat including powdery mildew, leaf, stem and stripe rust. The *Lr67/Yr46* originates from a Pakistani accession was transferred to 'Thatcher' (near-isogenic line RL6077; [203]). Later on, it was reported that the same locus confers resistance to stem rust (*Sr55*) and powdery mildew (*Pm46*) [204]. Recently, Ponce-Molina et al. [193] identified Pusa 876' (NP876) as a potential source for *Lr67/Yr46* and *Lr46/Yr29*. Chhetri et al. [205] also identified chromosome 4D as a pleiotropic locus for *Lr67/Yr46/Sr55* in W195/BTSS RIL population with at least two QTLs contributed by one or both of the parents for each trait.

The fourth pleiotropic APR gene is *Lr26/Yr9/Sr31/Pm8* involving 1B/1R translocation from Veery lines. Several Veery-derived varieties were developed and released in the 1980s and 1990s but became ephemeral thus increasing selection pressure on the pathogen variants virulent to *Pm8* [206]. Pleiotropic gene *Lr27/Yr30/Sr2/Pm?* on chromosome 3BS [207] may be another unidentified pleiotropic APR gene for PM. Aravindh et al. [208] pyramided several fungal resistance genes including *Sr36/Pm6*, *Sr2/Lr27/Yr30* and *Sr24/Lr24* in the same background.

## 6. Wheat Genetic Resources: Conservation and Use in PM Breeding Programs

### 6.1. Wheat Gene Banks as a Source of PM Resistance

Genetic diversity is the variability in one or few traits between organisms of the same species while genomic diversity was defined as the variability present at several gene-loci within an individual/organism [209]. Frequent use and repetitive planting of few parental lines/varieties across wide agro-ecosystems led to the erosion of genetic diversity thus limiting the improvement of wheat varieties. However, the differences in complex geographic regions, yearly variable climatic conditions, artificial and natural selection have contributed to the rich diversity of wheat germplasm sources [210]. The genetic improvement of wheat depends on the availability of adequate genetic diversity for agronomic, yield and quality performance and broad-spectrum resistance to disease and pest variants. Wheat genetic resources such as landraces, old varieties, and wild relatives are important sources of unexploited alleles and genes [170]. In view of the need to preserve the genetic diversity of crops, national and international gene banks have been established to preserve important genes to use in research and breeding programs aimed at genetic improvement. It has been indicated that around 7.4 million accessions are being preserved in 1700 genebanks worldwide [211].

The Consultative Group on International Agricultural Research (CGIAR) just marked its 50 years. Founded in 1971, the CGIAR is currently composed of 15 international agricultural research centers including the International Maize and Wheat Improvement Centre (CIMMYT) and International Center for Agricultural Research in the Dry Areas (ICARDA), collectively missioned to bring global research expertise and resources through the evaluation of wheat to adapt to different mega-environments such as high rainfall, irrigated, arid and semi-arid environments, high temperature, alkaline, saline, diseases and pests' prone environments for agricultural productivity growth, poverty alleviation and food security across the globe [212,213]. CIMMYT maintains the largest genebank and is mandated for providing germplasm to other wheat and maize improvement programs around the world. The prestigious genebank is the largest reservoir of accessions including wheat, maintaining nearly 200,000 entries. These genetic resources represent a wealth of untapped alleles and genetic diversity for potential exploitation in breeding programs. Despite the magnitude of these genebanks across the globe and the stored genetic resources, most of them are still underutilised. This is because the majority of germplasm have not been characterized for most important traits due to complex genetic profiles, limited availability of descriptors, lack of data regarding their taxonomy and geographic origin, loss of important alleles due to evolution and domestication process and the presence of deleterious alleles [214,215].

The full exploitation of genetic resources maintained in genebanks depends on the ability to effectively phenotype and genotype the accessions for resistance/tolerance biotic and abiotic stresses. Table 2 shows some of the leading national and international wheat gene banks. In South Africa (SA), the three main wheat improvement and germplasm maintenance centers include Syngenta in acquisition of Sensako, Pannar and Agricultural Research Council-Small Grain (ARC-SG), and to smaller extent, Monsanto in partnership with Grain SA. Of these, the ARC-SG is at the forefront of germplasm maintenance and wheat breeding in collaboration with educational institutions and private breeding companies. The ARC-SG currently holds more than 20,000 small grain accessions (for which most of them were imported from genebanks around the world) including wheat, oats, barley, rye and triticale, of which wheat accounts for nearly 90% of these collections [216]. However, less than 10% of these accessions have been tested for PM resistance (unpublished data). Furthermore an average of 50 accessions are distributed across the country i.e. universities, plant breeders, plant pathologists and entomologies [217] to test for abiotic and biotic stresses. However, no report has been made for testing for PM resistance. In the context of Sensako, all nine winter-rainfall adapted cultivars (SST's) are susceptible to powdery mildew in SA [217]. Recently, a study conducted revealed high virulence frequencies of South African PM isolates to *Pm6*, *Pm8*, *Pm17 Pm34* and *Pm35* [17].

**Table 2.** Important gene banks and databases of small grains, including wheat, as sources of PM resistance.

| Gene Bank | Institution or Country | Year of Establishment | Genebank Capacity | No. of Wheat Accessions | References/Website |
|---|---|---|---|---|---|
| The Consultative Group on International Agricultural Research (CGIAR, 15 centers) Genebank Platform | France | 1971 | ~770,000 accessions | - | CGIAR: Science for humanity's greatest challenges |
| Centre for Maize and Wheat Improvement (CIMMYT) | Mexico | 1966 | ~200,000 accessions | ~80,000 | https://www.cgiar.org/research/center/cimmyt/ |
| International Center for Agricultural Research in the Dry Areas (ICARDA) | Beirut, Lebanon | 1977 | ~150,000 accessions | - | ICARDA Annual report, 2021 |

**Table 2.** *Cont.*

| Gene Bank | Institution or Country | Year of Establishment | Genebank Capacity | No. of Wheat Accessions | References/Website |
|---|---|---|---|---|---|
| USDA—National Small Grains Collection (NSGC) or National Plant Germplasm System (NPGS) | Aberdeen, Idaho, USA | 1988 | ~143,893 accessions | - | https://www.ars.usda.gov/pacific-west-area/aberdeen-id/small-grains-and-potato-germplasm-research/docs/national-small-grains-collection/ and USDA-ARS-NPGS |
| Plant Gene Resources of Canada (PGRC) | Canada | 1970 | ~112,000 accessions | - | https://pgrc.agr.gc.ca/holdings-stocks_e.html |
| Grains Research and Development Corporation (GRDC) | Australia | 1990 | - | - | https://grdc.com.au/ |
| Institute of Plant Genetics and Crop Plant Research (IPK), Gatersleben | Germany | 1992 | ~150,000 accessions | ~22,000 | https://www.ipk-gatersleben.de/en/research/genebank |
| Genesys: Institute for Cereal Crops Improvement (ICCI) | Israel | 1970 | ~17,006 accessions | - | https://en-lifesci.tau.ac.il/icci |
| Pannar | South Africa | 1958 | | | |
| Agricultural Research Council–Small Grain (ARC-SG) | South Africa | 1976 | ~20,000 accessions | 17,551 | https://www.arc.agric.za/Documents/Annual%20Reports/AR2021-low%20res-OCT%202021.pdf |

### 6.2. Wheat Databases as a Source of PM Resistance

In the past few decades, wheat QTL analysis was conducted on diversity of individual traits, making available the linked markers, map or genomic positions and the contribution of the phenotypic variation of the traits of interest [115,181,218,219]. Recently, the lack of a completely sequenced reference genome in common wheat has limited the discovery of candidate genes/QTLs. However, the recent advancement in functional genomics has revolutionised the discovery of candidate genes/QTLs for the adaptation of lines to biotic and abiotic stresses. Recently, genome-wide association studies (GWAS) have made it possible to exploit linkage disequilibrium (LD) between tightly linked polymorphic markers and QTLs in a large number of germplasm. Nevertherless, extensive databases for curating wheat QTLs are still infant. To increase the competitiveness of public wheat breeding programs through the intensive use of modern selection technologies, mainly marker assisted selection (MAS), several databases have been developed. Few of those include MASwheat, GrainGenes, Wheat Expression browser and WheatQTLdb, whereby thousands of biotic and abiotic (stress, biofortification traits, morphological traits as well as yield and end-use quality traits genes, alleles and QTLs have been curated [220–222]. The Leibniz Institute of Plant Genetics and Crop Plant Research (IPK) Gatersleben, Germany is currently preparing a large Wheat data warehouse web portal for the same purpose (unpublished). These databases further provide access to various germplasm, gene expression, genome-specific primers, sequences, QTLs (metaQTLs and epistatic), as well as linked publications.

Established in 1992, Graingenes has curated data from various genera including *T. aestivum*, *Ae. tauschii*, *Avena sativa*, *Hordeum vulgare*, *Secale cereale*; indexing about 548 QTLs, 91 genetic maps, 10 physical maps, 14,411 germplasm records in collaboration with Wheat Information System (WheatIS) and 3119 genes in collaboration with Wheat Gene Catalogue [220]. In the context of WheatQTLdb, V1.0 and V2.0, were developed between 2020 and 2022 where V1.0 only focused on hexaploid wheat. The updated WheatQTLdb V2.0 has now expanded to provide information on wheat and its seven other related species including *T. durum*, *T. turgidum*, *T. dicoccoides*, *T. dicoccum*, *T. monococcum*, *T. boeoticum* and *Ae. tauschii*. Between V1.0 and V2.0, about 11,552 and 27,518 QTLs, 330 and 1321 metaQTLs and 107 and 202 epistatic QTLs were extracted and curated from wheat [221,222]. By 2022, about 3706 QTLs have been curated for fungal resistance [222].

This is the largest database serving curators, breeders, researchers, and geneticists with exhaustive information to use in fine mapping, association mapping, cloning and MAS.

### 6.3. Genetic Resources of Wheat for Powdery Mildew Resistance

### 6.3.1. Wheat Landraces

Landraces are the significant repository of a diverse gene pool due to the broad intraspecific genetic diversity and consequently contribute to sustainable agricultural practices [223]. Table 3 shows some landraces reported with PM resistance and other useful agronomic traits. Wheat landraces are genotypes with wider genetic diversity than improved or commercial cultivars which are more prone to stresses i.e. abiotic and biotic [224]. The genetic diversity of landraces is the foundation of stable and intermediate to higher yield levels even under low input agricultural systems, disease and pest resistance, excellent adaptation to changing climates (drought, heat and cold), and good agronomic wheat traits [9,225,226]. Therefore, valuable farm- and market-preferred breeding traits can be readily introduced from landraces into well-adapted and high-yielding wheat varieties to ensure food security [227].

Exploiting new genetic diversity in elite or novel genetic resources to produce suitable genotypes with broad-spectrum resistance to fungal diseases is still an ongoing ambition in wheat breeding programs. Several PM resistance genes have been derived from wheat landraces including *MlHLT* [168], *MlXBD* [170,228], *pmDGM* [172], *pmDHT* [229], *pmHYM* [171,229] *pmQ* [173], *pm* [191], *pmYBL* [230], *Pm3* [16], *Pm5* (231179), *Pm223899* [231], *Pm24* [133,232], *Pm45* [158], *Pm47* [152], *Pm59*, [158] *Pm61* [160] and *Pm63* [161]. Alleles for broad-spectrum PM resistance have been identified in Chinese landraces including *Pm2c* [233] *Pm3b* [24], *Pm5d* [120], *Pm5e* [121] and *Pm24b* [133]. Though a limited number of wheat landraces were reported resistant to PM isolates in the US (59 of 1 297), it is suggested that there is still hope in exploiting landraces for sought-after traits including PM resistance [19]. From different studies, it is evident that host plant resistance to wheat powdery mildew can be redesigned with landraces through the introgression of important resistance genes/alleles.

### 6.3.2. Tetraploid Wheats

Several important genes for biotic stress resistance have been transferred into common wheat from the primary gene pool of tetraploid wheat (*T. turgidum* ssp. *dicoccoides*, ssp. *dicoccum*, WEW, and ssp. durum, DW), such as those related to the most dreadful and economically important diseases of wheat. Durum wheat has been used as source of PM resistance genes (*Mld*, *MlIW72*, *Pm3h* and *PmDR147*) for bread wheat improvement [25,234–237]. *Mld* (4B, recessive) was employed in wheat breeding in combination with other *Pm* resistance genes, such as *Pm2* (5DS, [238]) and *Pm3h* (1AS, dominant, [236]), and probably originated from an Ethiopian durum wheat accession [25]. *PmDR147* (2AL, dominant) was transferred into bread wheat cv. 'Laizhou 953' from the durum wheat accession DR147 [239]. Two powdery mildew resistance genes, formally named *Pm5a* and *Pm4a*, identified in cultivated emmer, were used for bread wheat improvement. *Pm5a*, (7BL, recessive; [240]) appeared in the varieties 'Hope' and 'H-44' along with *Sr2*, while the dominant gene *Pm4a* (2AL, dominant; [241]), was transferred to bread wheat variety 'Chancellor' from the Indian emmer landrace 'Khapli' [242]. Wild emmer wheat (WEW, *T. turgidum* ssp) is a main source of PM resistance genes - twenty-one - for hexaploid wheat including *Pm16*, *Pm26*, *Pm30*, *Pm36*, *Pm41*, *Pm42* and *Pm64*, among others [139,145,148,189,243,244]. A direct transfer from WEW into bread wheat was done for 13 of them, while for the others an identification/mapping after a crossing with durum wheat or a validation/mapping in durum background, followed by transfer into hexaploid wheat was undertaken [245].

### 6.3.3. Synthetic Hexaploid Wheat

Compared to its donor species, the genetic diversity of bread wheat is narrow [246]. To enhance the effectiveness of genes, breeders have created a pathway of transferring genes from rye, einkorn and wild emmer wheat. Synthetic hexaploid wheat (SHW) is an artificially derived wheat with an eclectic hereditary base due to introduced and altered genetic fragments from wheat progenitors and wild relatives including tetraploid (*T. turgidum*), goatgrass (*A. tauschii*) and diploid wheat (*T. urartu*) [22,159,247]. The genomic interactions of the tetraploid and diploid resources may cause complex changes in the genetic, epigenetic and biochemical basis of the resulting SHW. Since the late 1800s and early 1900s, wheat and rye were successfully crossed to combine the traits of the two parents to form a new intergeneric hybrid. This was aimed at associating rye cold tolerance to several diseases and its adaptation to soil and climate conditions with the wheat productivity and nutritional qualities [248–250]. McFadden and Sears [251] successfully initiated artificial synthesis of hexaploid wheat with *T. turgidum* and *Ae. tauschii*. Ever since this success, synthetic hexaploids were acquired globally [252–254]. To date, more than half of PM resistance genes/QTLs were introgressed from wheat progenitors and wild relatives. Some examples are reported in Table 2.

Synthetic derivative lines (SDLs) have been recognised as major parents for conventional breeding with intense selection resulting in advanced lines with excellent yield performance and disease resistance [246,255]. SHW 'SE5785' has been a major source of wheat PM resistance gene *PmSE5785* located on chromosome 2D and thus SDLs N07228-1 and N07228-2, with large seeds and powdery mildew resistance, were selected from the 'SE5785'/'Xiaoyan 22' cross [256]. *Pm53* was introgressed from *Aegilops speltoides* into soft red winter wheat located on chromosome 5BL [184]. *Pm41*, *Pm42* and *Pm50* derived from wild emmer wheat (*T. turgidum* var. *dicoccoides*), are located on chromosome 3BL [148,149,257] and 2BL [151]. *Pm62* (introgressed from *Dasypyrum villosum*) and *Pm50* are 2 APR genes located on wheat chromosome arm 2VL [188] and 2AL, respectively [257]. *Pm60* was derived from diploid wild wheat (*T. urartu*) [30,159]. To date, hundreds of SDLs for numerous traits have been registered/released globally including China, Iran, Ethiopia, India, Kenya, Pakistan, Mexico, Turkmenistan, Turkey, Tajikistan, Syria, Morocco, Uruguay, Afghanistan, Argentina and Spain [255,258,259]. Of these countries, China has proved to value gene pool introduced from SHW as over 2000 SHW from the CIMMYT were introduced in the country since 1995. As a result, four SDL-cultivars (Chuanmai 38, Chuanmai 42, Chuanmai 43 and Chuanmai 47) have been released in China (222). Li et al. [246] reported that alleles from the four SHW-cultivars contribute to new traits such as resistance to stripe rust, pre-harvest sprouting and strong vegetative vigor, extra spikes per plant, additional grains per spike, superior grains, and higher grain-yield potential. Among the four cultivars, Chuanmai 42, released in 2003, has broken the yield record with great agronomic and quality traits (large grains of $\pm$ 50 g in 1000-grain weight and highest yield average of > 6 t/ha) and resistance to stripe rust. To date, the SDLs of the four cultivars are grown in more than 3,500,000 ha in south-western China. Recently, Chuanmai 104 (from parent Chuanmai 42, [260]) showed resistance to stripe rust and powdery mildew inheriting the resistance loci *QPm.saas-4AS* [261], *Qyr.saas-7B* [262] and *YrCH42* [263]. Therefore, it is evident that SHWs and SDLs through crossing with *T. aestivum* cultivars can eliminate deleterious traits or transfer the desirable traits [264–266].

Bi-parental breeding is a common approach used for breeding pure-lines in self-pollinated crops including wheat. A bi-parental approach is effectively used by researchers and plant breeders to identify superior parental lines from a candidate population to combine target traits before conducting extensive field trials [39,267]. In simple terms, the intention to improve both genetic diversity and selection efficiency and improve quantitative traits such as resistance to powdery mildew can be successfully attained by means of homozygous lines [22]. Multi-parental populations can be developed using the above-mentioned genotypes/genetic resources i.e. wheat landraces and synthetics as donor parents [93]. Furthermore, recombinant inbred lines (RILs) and doubled-haploid (DHs)

developed by crossing two inbred parental lines allow plant breeders to fix the desired combinations of genes/alleles/QTLs to produce lines with homozygous traits. The $F_1$ is selfed to produce the $F_2$ generation, and the subset of the two inbred parental lines from the $F_2$ generation is selected to produce the potential recombinations. The resulting combinations are usually called mapping populations intended for selecting/improving targeted genome(s) to map genes/QTLs that control the inheritance of resistance to powdery mildew hopefully at both the seedling and adult-plant stage [22,39,72].

Multiparent Advanced Generation Intercross (MAGIC) population represents intermediate to bi-parental crosses producing the gene pool with wider genetic and allelic diversity for a number of breeder/farmer preferred traits that can be explored further. MAGIC is prominent for allowing the high-resolution mapping of quantitative traits. In MAGIC population, multiple founder lines are selected based on superior traits (agromorphological and disease traits) and intercrossed several times in a well-defined order to combine the target quantitative traits of all the founder lines in a single line [268,269]. Since developing a cultivar may take up to ±7 years, this is different with a bi/multi-parental as it takes up to two years minimum. Genome-wide markers are also used to select the best progeny with desirable combinations [270]. The most widely used donor parents in developing mapping populations for powdery mildew resistance include Pingyuan 50 [78,271], Hongyoumai [171,229] and Baihulu [133] and Lumai 21 (LM21 [272,273]).

By selecting bi-parentals for crosses, breeders hope to generate progenies with a combination of favorable quantitative traits for superior performance and high yield stability under biotic and abiotic stresses [39]. However, every good comes with drawbacks. For example, the subset quantity generated from the parental lines often exceeds what can be handled by the breeders during screening either under a controlled environment or in the field. Furthermore, the truncation selection approach eliminates favorable alleles/genes/QTLs from the breeding population thus narrowing the genetic/genomic diversity [267]. Moreover, genotype-by-environment interaction presents one of the major challenges when conducting field trials using the subset of bi-parentals [14,274].

**Table 3.** Some modern wheat genotypes reporting PM resistance, agronomic, or other beneficial traits.

| Genotype Name | Type of Accession | Traits Type(s) or Gene | Country or Organization | Year of Release | References |
|---|---|---|---|---|---|
| Hongyoumai | Landrace | *pmHYM* | China | - | [171,229] |
| Duanganmang | Landrace | *PmDGM* | China | - | [172] |
| Baiyouyantiao | Landrace | *PmBYYT* | China | - | [210] |
| Xiaohongpi | Landrace | *pmX* | China | - | [191] |
| Pingyuan 50 | Landrace | *Powdery mildew* and *stripe rust* | | 1950s | [78,271] |
| Niaomai | Landrace | *Pm2c* | China | - | [233] |
| Hongyanglazi | Landrace | *Pm47* | China | - | [152] |
| Guizi 1 | Landrace | *PmGZ1* | China | - | [275] |
| Xiaobaidong and Fuzhuang 30 | Landrace | *mlxbd* and *mlfz* | Germany | - | [132,170,276] |
| Hulutou | Landrace | *MlHLT* | China | - | [168] |
| Xuxusanyuehuang 'XXSYH' | Landrace | *Pm61* | China | - | [160] |
| Baihulu | Landrace | *mlbhl* | China | - | [133,277] |
| Baihulu and Hulutou | Landrace | *Pm24* | China | - | [133,232] |
| Qingxinmai | Landrace | *PmQ* | China | - | [173] |
| Dahongtou | Landrace | *pmDHT* | China | - | [229] |
| Shangeda | Landrace | *PmSGD* | China | - | [278] |
| Youbailan | Landrace | *pmYBL* | China | - | [230] |
| Honghauaxiaomai | Landrace | *PmHHXM* | China | - | [279] |
| Dataumai | Landrace | *PmDTM* | China | - | [280] |

**Table 3.** *Cont.*

| Genotype Name | Type of Accession | Traits Type(s) or Gene | Country or Organization | Year of Release | References |
|---|---|---|---|---|---|
| Youzimai | Landrace | *Seedling resistance to powdery mildew* | China | - | [281] |
| PI 181356 | Landrace | *Pm59* | Great plains | - | [158] |
| PI 223899 | Landrace | *pm223899* | USDA-ARS, Oklahoma | - | [231] |
| PI 628024 | Landrace | *Pm63* | USDA-ARS, Oklahoma | - | [161] |
| Synthetic 43 | Synthetic | *pmT* | North Western Plain Zone of India | 1993 | [22] |
| SE5785 | SHW | *PmSE5785* | Chinese Academy of Agricultural Sciences, Beijing, China | - | [256] |
| N07228-1 and N07228-2 | SDL | Large seeds and PM resistance | College of Agronomy, Northwest A&F University, China | | [256] |
| Chuanmai 104 | SHW | APR to PM, stripe rust, and pre-harvest sprouting; high yielding, good quality, wide adaptability | Crop Research Institute, Sichuan Academy of Agricultural Sciences (CRI-SAAS) | 2012 | [246,261] |
| MG5323 | *T. turgidum* | *Ml5323* | University of Bari, Italy | | [135] |
| NC96BGTA4 | *T. monococcum* | Pm resistance | North Carolina Agricultural Research Service and the USDA-ARS | 1996 | [134] |
| NC96BGTA5 | *T. monococcum* | *Pm25* | North Carolina Agricultural Research Service and the USDA-ARS | 1996 | [134,282] |
| NC96BGTA6 | *T. monococcum* | PM resistance | North Carolina Agricultural Research Service and the USDA-ARS | 1996 | [134] |
| NC99BGTAG11 | *T. timopheevii* | *Pm37* | North Carolina Agricultural Research Service and the USDA-ARS | 2000 | [146,283] |
| MG29896 | *T. turgidum* | *Pm36*, high grain protein content, and acceptable seed size | University of Bari, Italy | - | [145] |
| Translocation line L50 | *Ae. speltoides* | *Pm32* | Technical university of Munich, Germany | - | [141] |
| Wild emmer IW2 | *T. dicoccum* | *Pm41* | Mount Hermon, Israel, | - | [148] |
| Wild emmer accession G-303-IM | *T. dicoccum* | *Pm42* | Israel | - | [149] |
| K2 | *T. dicoccum* | *Pm50* | Institute for Crop Science and Plant Breeding, Germany | - | [257] |
| CH7086 | Thinopyrum ponticum | *Pm51* | Crop Science Institute, Shanxi Academy of Agricultural Sciences | - | [154] |
| Qinling | *Secale cereale* | *Pm56* | Sichuan Agricultural University, Ya'an, China | - | [186] |
| NAU421 (T5VS·5AL) | *Dasypyrum villosum* | *Pm55* (growth-stage and tissue-specific dependent resistance) | Nanjing Agricultural University, China | - | [185] |
| TA1662 | *Ae. tauschii* | *Pm58* | Michigan State University, USA | - | [187] |
| T.urartu | *T. urartu* | *Pm60* | Jiangxi Normal University, China | - | [159] |

## 7. Breeding Methods and Technologies

### 7.1. Selection Using Phenotypic Traits: Classical Breeding

A substantial amount of research efforts have been invested in developing improved crop varieties through conventional breeding. This approach is the forefront of every plant research and breeding as it involves the act of variety improvement by informed breeding and selection of best-performing genotypes. This aims to develop and improve variety resistance to biotic and abiotic stresses, ensure resilient production and yield stability, increase profits and enhance global food security [9,284].

Conventional breeding has been the backbone of many breeding programs. This approach involves the use of natural germplasm collection, mapping/breeding populations using complementary genetic sources such as landraces, breeding lines, doubled-haploids (DHs), near-isogenic lines (NILs) and recombinant inbred lines (RILs) to deliver PM responsive traits, alleles, genes and QTLs. Hundreds/thousands of genotypes/accessions/families are mined for potential selection for PM resistance in wheat. The choice of the *B.gt* isolates is mostly based on their avirulence and virulence patterns to the known alleles/genes. This allows breeders to to screen a large set of germplsm with diverse *B.gt* isolates and select the promising lines, simultaneously reducing the sample size (discard the susceptible plants).

Phytopathological tests are carried out at seedling and adult plant stages under controlled and contrasting environments over a number of seasons [78,113,159,183,285–289]. This is done to phenotype complex disease traits including powdery mildew resistance, simultaneously assessing plant morphology, growth habit, plant height, grain yield and its contributing traits especially in the field under the target stresses [39,69,91,92,256]. These systems enable easier and quick differentiation of genotype reactions from the pathogen infections. For seedling studies, inoculations are performed by dusting conidia from infected seedlings to those under study and infection types (IT) are scored 8–12 days post-inoculation [133,183] using a scale from zero to four: highly resistant-resistant (IT = 0, 1), moderate resistance (IT = 2) and susceptible-highly susceptible (IT = 3 and 4). In the case of wheat powdery mildew APR, a disease index of 0–9 scale or 0–100% is used to measure and categorise genotype reactions (114 ,116). Genotypes reactions are usually classified into resistant, moderately resistant and susceptible. For durable resistance, genotypes with consistent performance over plant growth stages, environment and years are valuable in breeding programs (23). Evidently, the application of conventional breeding methods has significantly increased yields worldwide even under PM infestations. The most renowned success of conventional breeding is the semi-dwarf high-yielding cultivars developed during the Green revolution. Chuanmai 104 (CM104), is an elite SHW derived variety, with resistance to multiple traits i.e. powdery mildew, stripe rust, pre-harvest sprouting and low temperature; excellent agronomic traits i.e high yield and good quality as well as wide adaptability in China [246,261,290]. Therefore, the multi-trait resistance offered by Chuanmai 104 is valuable in breeding programs. Major success in breeding for resistance to wheat pothogens is attributed by *Pm* genes *Pm38*, *Pm39* and *Pm46*. The presence of these genes in a wheat variety/cultivar has made it easier to detect/identify the presence of genes for other pathogens including leaf rust, stem rust and stripe rust [179,181,193]. However, with the projections of human population growth and food demands by 2050, advanced breeding methods are needed to meet these future predictions. Thus, breeding programs should devise strategies such as breeding for or pyramiding high-yield, end-use quality traits and resistance to fungal pathogens in the same genetic background.

The major limitations of conventional breeding include the number of generations required for screening complex phenotypic traits under multiple environmental conditions and different seasons. This makes this approach labour intensive, time consuming and expensive [209]. Recording of the phenotypic data may also increase chances of errors in the measurement of the traits and the identification of false positive alleles. Estimating disease severity by visual assessment and scoring is very subjective and error-prone and in large scale screening, limits the efficiency and accuracy of phenotyping [23]. These bottlenecks have driven the development of high-throughput phenotyping platforms (HTPPs) relying

on automated imaging and the use of different sensors [95–99,291] and genome selection approaches/technologies, suitable for use in laboratories.

### 7.2. Marker-Assisted Selection (MAS)

Marker-assisted selection in plant breeding has become a common practice for the selection of traits with the aid of molecular markers. Of all known molecular markers types, diversity arrays technology (DArTs), single nucleotide polymorphisms (SNPs) simple sequence repeat (SSRs) are widely used in MAS [218,261,273,288]. This is because molecular markers are complementary tools to conventional breeding since they are highly heritable, easy to assay, faster, cheaper, more accurate and not affected by the environment. Furthermore, selecting of all traits of interest can be carried out at seedling stage thus reducing time required to phenotype [292]. To increase selection efficiency, a marker must be closely associated with the phenotype of interest. MAS enhances the selection of potential parental lines in breeding programs, elimination of bad linkage drag and selection of breeding traits that are difficult to measure using expensive and time-consuming phenotypic assays. Molecular markers also enable the characterization of varieties into what is referred to as distinctiveness, uniformity and stability (DUS) assessment, an association of alleles with traits of importance and inferences of population history.

To date, several MAS approaches have been successfully employed including foreground and background selection [61,293], also known as marker-assisted backcrossing (MABC) [294], linkage mapping [273,295], and mining or accumulation of favourable alleles in early generations [286,288,289,296], selection for quantitative APR for powdery mildew in wheat using GWAS [297], GWAS combined with genomic prediction and selection [298].

The transfer of important disease resistance genes/alleles/QTLs from closely related wheat species is often associated with bad linkage drag, however, such genes are often limited for commercialization. Furthermore, genes transferred from the wild relatives are often diluted/supressed in their resistance in the wheat background [35,36]. Therefore, foreground and background selection also known as marker-assisted backcrossing (MABC) can ensure that target genes are successfully transferred from wild and alien species into wheat with effective resistance genes and minimum linkage drag [293,299]. Gene pyramiding of multiple resistance conferring genes can be attained with these methods [61]. With these methods, resistance genes from an inferior source i.e. donor parent can be transferred into a recurrent parent i.e. well-adapted breeding cultivar or line [294]. In the case of MABC, the resultant progeny/generation are crossed to the recurrent parent and the cycle continues until a new line identical (>96% by BC$_4$) to the recurrent parent is generated, but with the target trait/gene from the donor parent [300]. Molecular markers closely linked to the target gene are what makes these methods faster, effective and successful [301]. For example, *Pm21* has been reported to confer broad-spectrum resistance to most *B.gt* isolates. To date, several wheat varieties containing *Pm21* including Lantian27, Jinhe9123, Nannong9918, Neimai836, Shimai14, Xingmai2, Yangmai18 and Yangmai21, among others [302–304] have been developed and cultivated on more than 3.4 million hectares since 2002 in China. Using MABC approach and *Pm21*-specific markers, high intensive selection resulted in the development of three wheat varieties Ningchun4, Ningchun47, and Ningchun50 *Pm21* resistance and post-flowering agronomic traits [305].

GWAS involves screening of markers across the organism's genome including wheat to identify genetic/genomic variations associated with complex diseases including powdery mildew. In the last decade, the phenomenon has greatly advanced the field of complex disease genetics such as PM in wheat through identifying novel significant and *bona fide* associations [295,297,306]. The GWAS approach overcomes the drawbacks elicited by bi-parental linkage mapping including restricted allelic diversity and limited genomic resolution [298]. However, the inability to illuminate the heritability of all the complex traits presents one of the major limitations of GWAS [306].

GWAS and genomic selection (GS) have been used in combination for stress tolerance and related traits, accelerating knowledge and understanding of genetic makeup under-

lying target-responsive traits for improvement in wheat [307]. However, the inadequate marker density presents one of the major limitations of the utility of GWAS and GS in wheat genomic breeding. With the constant decline in genotyping cost and increasing SNPs and DArT marker assay platforms, advances in genomic prediction and selection has allowed the use of large phenotypic and genetic diversity panels, revolutionising the field of plant-genomic breeding. The application of this approach increases the rate of genetic gain per unit price simultaneously reducing the length of breeding cycle [308]. GWAS, linkage mapping and genotype by sequencing have been successfully applied in genomic prediction studies to identify genes/QTLs associated with target traits [295,307,309]. Furthermore, set of diverse population i.e. bi-parental (DH and RILs), multi-parental, breeding lines, cultivars and landraces have been used in genomic prediction studies [298,309]. This is because genome prediction or selection captures all minor effect QTLs and identifies individuals with high genomic estimated breeding values (GEBV) for target traits [310], thus reducing the number of generations required to predict superior phenotypes. Of note, due to the parental background (RILs), transgressive segregation produces progenies with greater phenotypic diversity that exceed their donor parents while the genetic diversity is often limited [311].

Though MAS may be more advanced, its application in breeding programs is hindered by the following challenges/drawbacks: (1) not all markers are breeder-friendly, (2) false selection during recombination between the trait/gene/QTL of interest and the markers may occur, (3) QTL position or location may be incorrectly estimated, (4) most breeding programs are not trained to use MAS techniques thus lack understanding for implementation, (5) most breeding programs are not equipped with facilities and equipment's for carrying out MAS and (6) MAS may be expensive especially during sequencing [292].

## 8. Quantitative Trait Loci (QTLs) for Resistance to Wheat Powdery Mildew

Quantitative resistance has been linked with non-race-specific resistance, exhibiting polygenic resistance. This resistance type is usually associated with a durable resistance, partial resistance, slow mildewing or delayed infection, development and reproduction of the fungus and is quite observed at adult stage of the plant [312,313]. Quantitative trait loci (QTL) analysis employs molecular markers to study the genetic diversity or variation, to localise the genetic variants underlying the phenotype response of quantitative traits, their effects and interaction [314]. The phenomenon is among the intensive genetic breeding approaches adopted in large mapping populations to explore the genetic nature, pattern, magnitude, degree, and extent of genomic regions and genes enabling resistance to diseases. This approach on a genomic level has been successful through targeting stable QTLs in distinct environments with the aid of high-throughput, robust and diagnostic molecular markers. Several QTLs for powdery mildew resistance have been located using molecular markers [63,78,218,315–318].

The APR from cultivar Massey, Knox and Pingyuan 50 have shown durability against powdery mildew for decades [78,271,319,320]. QTLs for APR have been mapped, derived from many resistance sources including Forno [118], RE714 [317], Massey [319], Lumai 21 [63,272], Bainong 64 [63,321]. Even better, QTLs for APR have also been pyramided by crossing two cultivars Bainong 64/Lumai 21 with good agronomic traits and APR to *B.gt* (63) and Pingyuan 50/Mingxian 169 with PM and leaf rust resistance, Libellula with stripe rust and PM resistance [318,322]. Using RIL population derived from a cross between PuBing 3228 (P3228) and Gao 8901, QTL *QPm.cas-7D* for APR contributed by P3228 explained 64.44% of phenotypic variance [219]. For the past 6 decades, Pingyuan 50 has shown durable APR to powdery mildew. Using DH populations derived from Pingyuan 50/Mingxian 169, three QTLs *QPm.caas-2BS.2, QPm.caas-3BS and QPm.caas-5AL* were mapped on chromosomes 2BS, 3BS and 5AL, each contributing 5.3%, 10.2% and 9.1% of phenotypic variance [78]. The use of molecular markers has made it easier to locate the APR genes/QTLs across the wheat chromosomes and to estimate the additive effect of each gene. Marker *Xbarc13* associated with *Pm5055* gene was also associated with QTL

*QPm.caas-2BS.2* [78,323]. It is evident that such a molecular marker has the potential for effective use in MAS and gene pyramiding for APR resistance. Previously mapped QTLs *QPm.caas-2DS* and *QPm.caas-4BL.1* for stripe rust were identified in the same position for PM resistance while QTL *QPm.caas-7DS* from Libellula was located in the same lucus as *Lr34/Yr18/Pm38* [318]. Six QTLs for APR to PM were detected across environments including *QPm.heau-1BL* (coinciding in the same locus as *Yr29/Lr46/Pm39*), *QPm.heau-1DL*, *QPm.heau-2DL*, *QPm.heau-4BL*, *QPm.heau-5BL*, and *QPm.heau-6BS*. *QPm.heau-1DL* [218]. From all these findings, it is evident that each slow-mildewing/APR QTL has a different phenotypic effect and different QTL expressing post interaction with the pathogen and the environment. Furthermore, these results revealed that PM is quantitatively inherited. Thus, a combination of minor genes underlying such resistance can result in high levels of resistance. Therefore, understanding the mechanisms of quantitative resistance involved in wheat-powdery mildew interaction and using diagnostic, robust and high-throughput molecular markers for detecting the genes/QTL involved in APR is of paramount importance. Table 4 presents the summary of QTLs for resistance to wheat powdery mildew.

**Table 4.** Summary of reported quantitative trait loci (QTL) for resistance to wheat powdery mildew.

| QTL (s) | Chromosome | Donor | Reference |
|---|---|---|---|
| *QPm.caas-1A* | 1AL | Bainong 64 | [63,321] |
| *QPm.sfr-1A* | 1AL | Oberkulmer | [118] |
| *QPm.caas-1AS* | 1AS | Fukuho-komugi | [285] |
| *QPm.vt-1B* | 1B | Massey | [319,324] |
| *Qaprpm.cgb-1B* | 1B | Hanxuan 10 | [325] |
| *QPm.heau-1BL* | 1BL | Francolin#1 | [218] |
| *Lr46/Yr29/Pm39* | 1BL | Saar | [181] |
| *QPmAPR.lfl-1BL* | 1BL | Atlantis | [316] |
| *QPm.vt-1BL* | 1BL | USG 3209 | [324] |
| *QPm.caas-1BL.1* | 1BL | Zhou8425B | [315] |
| *QPm.sfr-1B* | 1BS | Forno | [118] |
| *QPm.heau-1DL* | 1DL | Francolin#1 | [218] |
| *QPm.sfr-1D* | 1DL | Forno | [118] |
| *QPm.icg-1D* | 1DS | Kinelskaya 60 | [114] |
| *QPm.inra-1D.1* | 1DS | RE9001 | [326] |
| *QPm.vt-2A* | 2A | Massey | [319,324] |
| *QPm.vt-2AL* | 2AL | USG 3209 | [324] |
| *QPM.sdau-2A* | 2A | Lumai 21 (LM21) | [273] |
| *QPm.sfr-2A* | 2AS | Oberkulmer | [118] |
| *QPm.vt-2B* | 2B | Massey | [319,324] |
| *QPm.inra.2B* | 2B | RE9001 | [326] |
| *Qaprpm.cgb-2B* | 2B | Hanxuan 10 | [325] |
| *QPm.sdau-2B* | 2B | Shannong "SN0431" | [273] |
| *QPm.caas2BL* | 2BL | Lumai 21 | [63,321] |
| *QPmAPR.lfl-2BL* | 2BL | Line 6037 | [316] |
| *QPm.vt-2BL* | 2BL | USG 3209 | [324] |
| *QPm.caas-2B* | 2BL | Fukuho-komugi | [285] |
| *QPm.uga-2BL* | 2BL | 26R61 | [156] |
| *QPm.inra-2B* | 2BL | RE9001 | [326] |
| *QPm.caas-2BS* | 2BS | Lumai 21 | [63,321] |
| *QPm.caas-2BS.2* | 2BS | Pingyuan 50 | [78] |
| *QPm.umb-2BS* | 2BS | Folke | [327] |
| *QPm.umb-2DL* | 2DL | Folke | [327] |
| *QPm.caas-2DL* | 2DL | Lumai 21 | [64,321] |
| *QPm.umb-2DL* | 2DL | Folke | [327] |
| *QPm.sfr-2D* | 2DL | Oberkulmer | [118] |
| *QPm.caas-2DS* | 2DS | Libellula | [322] |
| *QPm.inra-2D-a* | 2DS | RE9001 | [218] |
| *QPm.inra-2D-b* | 2DS | RE9001 | [118] |

**Table 4.** *Cont.*

| QTL (s) | Chromosome | Donor | Reference |
|---------|-----------|-------|-----------|
| *QPm.caas-3BL* | 3BL | Mingxian 169 | [78] |
| *Qaprpm.cgb-3A* | 3B | Hanxuan 10 | [325] |
| *QPm.nuls-3AS* | 3AS | Saar | [181] |
| *QPm.caas-3BS* | 3BS | Pingyuan 50 | [78] |
| *QPm.caas-3BS* | 3BS | Zhou8425B | [315] |
| *QPm.sfr-3D* | 3DS | Oberkulmer | [118] |
| *QPm.tut-4A* | 4A | Line 8.1 | [116] |
| *QPm.uga-4A* | 4A | AGS 2000 | [156] |
| *QPm.sfr-4A.1* | 4AL | Forno | [118] |
| *QPm.sfr-4A.2* | 4AL | Forno | [118] |
| *QPm.caas-4BL.1* | 4B | Libellula | [322] |
| *QPm.heau-4BL* | 4BL | Francolin#1 | [218] |
| *QPm.sfr-4B* | 4BL | Forno | [118] |
| *QPm.caas-4BL.2* | 4BL | Zhou8425B | [315] |
| *QPm.saas-4AS* | 4BS | Chuanmai104 (CM104 | [261] |
| *QTL qApr4D* | 4D | Huapei 3 | [328] |
| *QPm.caas-4DL* | 4DL | Bainong 64 | [63,321] |
| *QPm.sfr-4D* | 4DL | Forno | [118] |
| *QPm.caas-5AL* | 5AL | Pingyuan 50 | [78] |
| *QPm.nuls-5A* | 5AL | Saar | [181] |
| *QPm.umb-5AL* | 5AL | Folke | [327] |
| *QPm.sfr-5A.2* | 5AL | Oberkulmer | [118] |
| *QPm.sfr-5A.3* | 5AL | Oberkulmer | [118] |
| *QPm.icg-5A* | 5AS | Kinelskaya 60 | [114] |
| *QPm.heau-5BL* | 5BL | Francolin#1 | [218] |
| *QPm.sfr-5B* | 5BL | Oberkulmer | [118] |
| *QPm.umb-5BS* | 5BS | Folke | [327] |
| *QPm.nuls-5B* | 5BS | Saar | [181] |
| *QPmyz.caas-5DS* | 5BS | Yangmai 16 | [329] |
| *QPm.inra-5D* | 5D | RE714 | [317] |
| *QPm.inra6A2* | 6A | RE714 | [317] |
| *QPm.icg-6A* | 6AL | Kinelskaya 60 | [114] |
| *Qaprpm.cgb-6B* | 6B | Hanxuan 10 | [325] |
| *QPm.uga-6BL* | 6BL | AGS 2000 | [156] |
| *QPm.caas-6BL.1* | 6BL | Huixianhong | [318] |
| *QPm.caas-6BL.2* | 6BL | Huixianhong | [318] |
| *QPmyz.caas-6BL* | 6BL | Zhongmai 895 | [329] |
| *QPm.caas-6BS* | 6BS | Bainong 64 | [321] |
| *QPm.sfr-6B* | 6BS | Forno | [118] |
| *QPm.umb-6BS* | 6BS | Folke | [327] |
| *QPm.caas-6BS* | 6BS | Bainong 64 | [321] |
| *QPm.caas-7A* | 7A | Bainong 64 | [321] |
| *Qaprpm.cgb-7A* | 7A | Hanxuan 10 | [325] |
| *QPm.sfr-7B.1* | 7BL | Forno | [118] |
| *QPm.sfr-7B.2* | 7BL | Forno | [118] |
| *QPm.nuls-7BL* | 7BL | Saar | [181] |
| *QPmyz.caas-7BS* | 7BS | Zhongmai 895 | [329] |
| *QPm.caas-7DS* | 7D | Libellula | [318] |
| *Qaprpm.cgb-7D* | 7D | Hanxuan 10 | [325] |
| *Lr34/Yr18/Pm38* | 7DS | Saar | [181] |
| *QPm.caas - 7DS* | 7DS | Chinese Spring | [315] |

## 9. Conclusions and Outlook

Powdery mildew is one of the most economically important diseases affecting wheat production. Chemical control methods for powdery mildew are expensive and pose hazards to humans and the environment. Thus, integrating host-plant resistance has been considered to be a sustainable and environmentally friendly option to control the

disease. Developing powdery mildew-resistant cultivars depends on identifying suitable sources of resistance and their effective implementation into breeding programs. More than 240 genes, including alleles, have been reported for resistance to wheat PM. However, most of these genes have been derived from wild relatives of wheat, limiting their commercial deployment owing to linkage drag and association with deleterious genes. The lack of precision and low selection efficiency for powdery mildew resistance using conventional breeding methods has resulted in limited success. The environmental variance, non-durable PM resistance, and polygenic nature of PM resistance have contributed to poor progress in PM resistance breeding. As demand for wheat grows rapidly across the globe, new breeding strategies, technologies, and tools are being used to urgently address the challenges associated with biotic and abiotic stresses such as growing climatic change, pests, and diseases that hinder domestic wheat production. The advent of high-throughput phenotyping, genotyping, and phenomics approaches holds the promise of improving selection efficiency and can be used to complement conventional breeding methods.

**Author Contributions:** T.B.: Conceptualization, original draft preparation, and editing. H.S.: supervision, project administration, content contribution, and editing. T.J.T.: supervision, project administration, content contribution, and editing. T.T.: content contribution and editing. S.B.: content contribution and editing. J.S.-M.: content contribution and editing. D.D.: content contribution and editing, F.D.: content contribution and editing. All authors have read and agreed to the published version of the manuscript.

**Funding:** The Winter Cereal Trust and AgriSETA/South Africa are thanked for bursary support to the first author. The Agricultural Research Council–Small Grain Institute, the University of KwaZulu-Natal/South Africa, and the National Research Foundation are acknowledged for the overall research support. JSM is a recipient of the grant "Ramon y Cajal" Fellowship RYC2021-032699-I funded by MCIN/AEI/10.13039/501100011033 and by the "European Union NextGenerationEU/PRTR". JSM acknowledges the support of the Junta de Castilla y León through the projects "Escalera de Excelencia CLU-2018-04, and CL-EI-2021-04 support to the internationalization of AGRIENVIRONMENT—Unidad Producción Agrícola y Medioambiente" of the University of Salamanca, both co-financed by the European Regional Development Fund (ERDF "Europe drives our growth"). SB is the recipient of the Swedish Research Council for Sustainable Development (FORMAS) Early-Career Researchers Grant number: 2020-01007. FD is a recipient of the PRIMA project CEREALMED "Enhancing diversity in Mediterranean cereal farming systems" (2020–2022). SusCrop-ERA-NET (2023–2025) WheatSecutity.

**Data Availability Statement:** Not applicable.

**Acknowledgments:** This work is supported by the Agricultural Research Council–Small Grain Institute. The authors greatly appreciate the input from all reviewers.

**Conflicts of Interest:** The authors declare no conflict of interest.

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
