# Peer review of "Breeding Wheat for Powdery Mildew Resistance: Genetic Resources and Methodologies—A Review"

_agronomy, doi:10.3390/agronomy13041173_

Round 1

Reviewer 1 Report

The purpose of this review is to present an overview of the current state of breeding for powdery mildew (PM) resistance, including the pathogenesis and distribution of PM isolates, key genebanks and databases, available genetic resources, and complementary breeding strategies for developing cultivars that are resistant to powdery mildew. Additionally, the review aims to provide insights into future developments in this field.

Comments:

Sections: 1. Introduction and 2. Constraints to wheat production

These sections need to be improved to explain the importance to achieve more efficient and long-lasting resistance, it is necessary to identify or create new sources of resistance against emerging pathogen strains. Why is it crucial to replace cultivars that lack effectiveness and diversify the sources of resistance by incorporating multiple resistance genes?

Section: 3. Pathogenesis, distribution and economic importance. It is well written and covered the most aspects of pathogenesis including life cycle and epidemiology, damage, and population genetics. However, more details are required to cover the second part of this section “distribution and economic importance”. Also, in deep explanation of Figure 1 is required to be included in the text.

Section: 4. Current control strategies. It is the most important section in this review. However, I recommend the authors to give more attention to” monitoring: Remote sensing technologies” which should be detailed and explained.

Section: 5.1 Resistance types for powdery mildew. Information related to Race-specific resistance and Non-race specific resistance should be summarized in a table.

Section: 7 Breeding methods and technologies includes 2 subsections of Classical breeding versus marker-assisted selection. However, the main aspects of comparison between these strategies are not well explained. Also, a confusion is raised with section 8. Quantitative trait loci (QTLs) for adult plant resistance.

Sections 9. Transgenic PM resistance in wheat and 10. Genome Editing in Wheat should be removed from this review. These section should be covered in more details in a way that can not be included in this review. So, I recommend the authors to remove these 2 sections and put more explanation for breeding methods and technologies.

Reviewer 2 Report

The manuscript is well-written and well-presented and should be of interest to your readers. Overall, provides a comprehensive analysis of current research on breeding wheat for powdery mildew resistance, however, the length of the manuscript makes it difficult to follow at times, and editing down some sections would greatly improve its readability. In addition, I have identified some errors in the text and provided comments that may be useful.

L2-3 - It is recommended to include a reference to the fact that this article is a review of the current state of the art in the title, as this information is currently only mentioned in the abstract.

L21 and Section 3 - The text contains numerous hyphens that are incorrectly separating words, likely as a result of copy-pasting formatting. Please review and correct the entire text accordingly.

L23 - QTLs is an acronym and it is only properly identified in Section 8.

L44 - It is advisable to avoid repeating the same words in both the title and the keywords, as this can decrease the visibility of the article. In particular, using the exact same keyword "wheat" as in the title may not be the best choice, and the authors could consider an alternative.

L61… - Italic formatting is missing from "et al.".  Check through the entire manuscript and ensure that the rest of the text is also properly formatted.

L77-80 - “Wheat (sensu lato) can be infected by B.g. tritici (hexaploid bread wheat) B.g. dicocci (tetraploid durum wheat) as well as B.g. triticale which is a hybrid between wheat and rye mildew with an expanded host range can infect triticale, and wheat (Menardo et al., 2016).” The writing and punctuation of this sentence need to be revised, as the current form of the sentence renders the final portion unclear.

L99 - Again, SA is an acronym, and it is only properly identified in the line 658, in section 6.1.

L100 - It's important to properly identify what Pm genes are.

L158 - “Dense cultivation associated with the use of semi-dwarf and high levels…” Replace with “semi-dwarf wheat varieties” to make the meaning of the sentence clearer.

L175 - “From the 346 isolates…” of what? Isolates should be identified to make the sentence clearer.

L194 - “… 3 to 5 days of free water and moisture contact/exposure.” Review and correct the meaning of this sentence since in its current form, the final portion does not make much sense.

L195 - “The secondary infections…” Are you referring to asexual reproduction?

L217 - Adverse instead of “…advere…”

L224- 236 - This portion of text should be part of the caption for Figure1. Also, only Fig. 1G is referenced in the text, and the remaining figures, Fig. 1A to F and H, are missing references.

L315 - Preventive instead of “…pre-emptive…”

L679-683 - The beginning of these two sentences, in this portion of the text, should be reviewed and changed.

Round 2

Reviewer 1 Report

The revised version was improved, and the authors addressed most of my comments. However, the authors did not clarify my comments concerning to "Information related to Race-specific resistance and Non-race specific resistance should be summarized in a table."  Table 1 is a text and not acceptable, it should be converted into a data records such as all others tables in the review.
